# Theaflavin-Enriched Fraction Stimulates Adipogenesis in Human Subcutaneous Fat Cells

**DOI:** 10.3390/ijms20082034

**Published:** 2019-04-25

**Authors:** Phil June Park, Chan-Su Rha, Sung Tae Kim

**Affiliations:** 1Basic Research & Innovation Research Institute, AmorePacific Corporation R&D Unit., 1920, Yonggu-daero, Giheung-gu, Yongin-si, Gyeonggi-do 17074, Korea; 2Vital Beautie Research Institute, AmorePacific Corporation R&D Unit, 1920, Yonggu-daero, Giheung-gu, Yongin-si, Gyeonggi-do 17074, Korea; teaman@amorepacific.com; 3Department of Pharmaceutical Engineering, Inje University, Gimhae-si 50834, Korea

**Keywords:** anti-aging, adipogenesis, green tea, human subcutaneous fat cells, theaflavin

## Abstract

Skin provides the first defense line against the environment while preserving physiological homeostasis. Subcutaneous tissues including fat depots that are important for maintaining skin structure and alleviating senescence are altered during aging. This study investigated whether theaflavin (TF) in green tea (GT) has skin rejuvenation effects. Specifically, we examined whether high ratio of TF contents can induce the subcutaneous adipogenesis supporting skin structure by modulating lipid metabolism. The co-fermented GT (CoF-GT) fraction containing a high level of TF was obtained by co-fermentation with garland chrysanthemum (*Chrysanthemum coronarium*) and the conventionally fermented GT (F-GT) fraction was also obtained. The effects of the CoF- or F-GT fractions on adipogenesis were assessed using primary human subcutaneous fat cells (hSCF). Adipogenesis was evaluated based on lipid droplet (LD) formation, as visualized by Oil Red O staining; by analyzing of adipogenesis-related factors by real-time quantitative polyperase chain reaction (RT-qPCR); and by measuring the concentration of adiponectin released into the culture medium by enzyme-linked immunosorbent assay. TF-enriched CoF-GT fraction did not adversely affect hSCF cell viability but induced their adipogenic differentiation, as evidenced by LD formation, upregulation of adipogenesis-related genes, and adiponectin secretion. TF and TF-enriched CoF-GT fraction promoted differentiation of hSCFs and can therefore be used as an ingredient in rejuvenating agents.

## 1. Introduction

Skin senescence is caused by intrinsic and extrinsic factors and leads to a loss of integrity and physiological functions of skin [1]. A decline in skin stiffness is a characteristic of aging [2]. Intrinsic factors related to skin aging include ethnicity (e.g., pigmentation), anatomical variations, and hormonal changes; extrinsic factors include environment (e.g., temperature and humidity), lifestyle (e.g., smoking/nicotine intake), and exposure to sunlight (e.g., ultraviolet radiation, UVR) [3,4]. Healthy skin maintains homeostasis and metabolic functions through communication among dermal cells such as keratinocytes, fibroblasts, melanocytes, and subcutaneous fat cells [5].

Subcutaneous adipose tissue plays an important role in skin rejuvenation [6]; subcutaneous adipocytes interact with fibroblasts and associate with elastic fibers in the dermal layer, thereby influencing the mechanical and structural properties of skin layers [7]. However, these fat-storing cells become thinner with aging and show a reduction in thermogenic capacity and structural stability including dermal elasticity, leading to skin wrinkling [8,9]. For example, UVR-induced photo-aging can modulate lipid metabolism, leading to reduced free fatty acid and triglyceride (TG) levels in adipocytes [10]. Therefore, the decreased function of adipocytes influences lipid metabolism in skin and cellular uptake of circulating free fatty acids, which can cause adverse health outcomes such as dyslipidemia [11], metabolic syndrome [12], and insulin resistance [13].

Green tea (GT) made from *Camellia sinensis* leaves is a widely consumed beverage that contains polyphenolic compounds with various health benefits [14] such as catechins, theaflavins (TFs), and thearubigins [15], whose abundance may be altered during the fermentation process [16]. TFs are known to influence lipid metabolism [17,18,19]. There are four major types of TFs—i.e., theaflavin (TF), theaflavin-3-gallate (TF3G), theaflavin-3′-gallate (TF3′G), and theaflavin-3,3′-digallate (TFDG)—that are produced in vitro from fresh tea leaves through oxidation by polyphenol oxidase; additionally, tea catechins in black tea leaves are generated by horseradish peroxidase [20,21]. Black tea polyphenols including TFs were found to have an anti-obesity effect in a mouse model [19]. Consumption of black tea prevents fat storage in liver, lowers lipid as well as glucose levels, increases fecal excretion of TGs, and diminishes adipose tissue [22]. TFs were also shown to block UV-induced skin cancer by suppressing UVB-induced activator protein-1 activity via inhibition of extracellular signal-regulated protein kinase and c-Jun NH2-terminal kinase [23]. TFDG has anti-melanogenic effects that are exerted via downregulation of tyrosinase protein and mRNA levels [24]. Black tea extract containing TFs may be used as a skin-lightening agent in the cosmetic industry owing to its anti-melanogenic effect [25]. Therefore, TFs have both anti-obesity and hypopigmentation-inducing properties. However, although TFs have been linked to lipid metabolism, their effects on subcutaneous tissue—and particularly adipocytes—are not known.

To address this issue, the present study was conducted to investigate the potential anti-aging effects of TFs. TF-enriched co-fermented GT extract (CoF-GT) was obtained from natural GT by co-fermentation with garland chrysanthemum (GC; *Chrysanthemum coronarium*) to increase TF concentration, and its effects were evaluated in cultured human subcutaneous fat cells (hSCF).

## 2. Results

### 2.1. Preparation of Theaflavin (TF)-Enriched Fraction by Co-Fermentation with Chrysanthemum Coronarium (GC)

Polyphenols in tea leaves are altered during the transformation from GT to black tea by fermentation via enzymatic oxidation [21,26]. Secondary polyphenols are generated through this process, with some changing the color of tea to brown [15,27]. To achieve high concentrations of TF, we prepared a TF-enriched fraction by co-fermentation with GC, and changes in the components were assessed by high-performance liquid chromatography (HPLC) (Figure 1). It was first confirmed that the four major TFs were well-isolated and showed different retention times after injection of TFs as standard chemicals (Figure 1A), comparing to the GT extract (Figure 1B). Then, it was verified whether CoF- or F-GT had different composition of polyphenols. As result, the content ratio of TF was higher than that of other components in CoF-GT (Figure 1C). F-GT fermented in a conventional manner without GC showed similarly increases for the four major TFs, without a significant change in overall TF content (Figure 1D and Appendix A). CoF- and F-GT fractions obtained in the preceding fermentation step were used for subsequent experiments. In addition, representative polyphenols such as TF and TFDG were also used as a positive control for a better understanding of CoF- and F-GT effects.

### 2.2. TF-Enriched Fraction Does Not Adversely Affect Human Subcutaneous Fat Cells (hSCF) Cell Viability

We next investigated the effects of TFs on cell proliferation and viability by treating undifferentiated hSCF cells with CoF-GT, F-GT, TF, or TFDG at concentrations ranging from 1 to 5 μg/mL for 24 and 72 h (Figure 2). After 24 h, cell proliferation was increased in the CoF- and F-GT groups relative to the TF and TFDG groups (Figure 2A). After 72 h, there was no evidence of cytotoxicity in any group and in some cases cell viability was slightly increased (Figure 2B). Therefore, polyphenols in CoF- and F-GT extracts promote hSCF cell growth and are safe for human cells.

### 2.3. TF-Enriched Fraction Induces Lipogenesis in hSCF Cells

Pre-adipocytes and mature adipocytes were prepared from hSCF cells. The differentiation of hSCF cells was confirmed by the formation of intracellular lipid droplets after 21 days of culture (Figure 3A, lower left panel) because lipid droplet formation is closely related to adipocyte differentiation and its observation could be useful for monitoring the differentiation. The amount of intracellular lipid in mature hSCF cells was evaluated by Oil Red O staining (Figure 3A,B) and by analysis of triglyceride (TG) content (Figure 3C). We examined the effect of TF and TFDG on hSCF cell differentiation and found that TF at concentrations ranging from 0.5 to 2 µg/mL increased the abundance of intracellular lipid droplets (Figure 3A, the right upper panel). Similar results were obtained for TG content. TFDG had the opposite effects in the same concentration range. Therefore, TF induces whereas TFDG inhibits lipogenesis in hSCF cells.

To assess the effects of CoF- and F-GT on hSCF cell differentiation, cells were treated with the test substances at concentrations ranging from 1 to 10 μg/mL for 21 days (Figure 3B). Treatment with 10 μg/mL CoF-GT induced lipogenesis (Figure 3B, upper panel), while the same concentration of F-GT had the opposite effect. We analyzed the content of TGs, a major component of lipid droplets, and found that TF (2 μg/mL) and CoF-GT (10 μg/mL) increased whereas TFDG (2 μg/mL) and F-GT (10 μg/mL) decreased TG levels (Figure 3C). The latter effect was accompanied by inhibition of hSCF cell differentiation. These results indicate that TF and TF-enriched fractions (e.g., CoF-GT) stimulate the lipogenic differentiation of hSCF cells.

### 2.4. TF-Enriched Fraction Increases Expression of Lipogenesis-Related Genes in hSCF Cells

We measured the mRNA expression of lipogenesis-related genes including *peroxisome proliferator activated receptor gamma* (PPARγ) and *adiponectin* (ADIPOQ) by real-time quantitative polyperase chain reaction (RT-qPCR) (Figure 4A). The transcript level of PPARγ, a master regulator of adipogenesis, was about six times higher in mature as compared to undifferentiated hSCFs after 21 days of differentiation (Figure 4A, black bar). TF treatment increased PPARγ expression in a dose-dependent manner; however, the level was decreased and reached a minimum value upon treatment with 2 μg/mL TFDG. In the CoF-GT group, PPARγ gene expression showed a dose-dependent increase, which is in accordance with the ORO staining results (Figure 3B). Furthermore, PPARγ expression was higher in cells treated with 10 μg/mL CoF-GT than in those treated with 2 μg/mL TF, and was downregulated in a dose-dependent manner in the F-GT treatment group.

The expression of ADIPOQ, a mature adipocyte marker [28], was four times higher in mature as compared to undifferentiated hSCF cells (Figure 4A, white bar). ADIPOQ level was increased by 2 μg/mL TF and decreased by 2 μg/mL TFDG treatment. Overall, the changes in ADIPOQ expression in the presence of TF and TFDG were similar to those observed for PPARγ, and were in agreement with the histological findings.

### 2.5. TF-Enriched Fraction Stimulates Adiponectin Secretion in hSCFs

Adiponectin (APN) is one of the adipocyte-secreted cytokine (adipokine) that is synthesized in LDs. The amount of APN in the culture media was higher for differentiated cells than for pre-adipocytes (Figure 4B). APN level was increased by treatment with TF (2 μg/mL) and CoF-GT (10 μg/mL) and reduced by TFDG (2 μg/mL) and F-GT (10 μg/mL), exhibiting a trend comparable to that of ADIPOQ mRNA expression in the RT-qPCR analysis. These results confirm that TF and CoF-GT promote lipogenesis in hSCF cells.

Based on the above findings, we propose the following model for the role of TFs in skin aging. TF and TF-enriched CoF-GT fraction promotes the differentiation of hSCF cells whereas conventionally fermented F-GT fraction has the opposite effect, as evidenced by up- and downregulation, respectively, of *PPARγ* and *ADIPOQ* expression. The increase in adipocyte marker gene expression was accompanied by intracellular LD formation. Therefore, TF enhances whereas TFDG inhibits hSCF differentiation. Additionally, although TF-enriched CoF-GT stimulated adipogenic differentiation, there were no changes in polyphenol content. Therefore, polyphenol content ratio varies according to the fermentation conditions and is the main factor regulating hSCF differentiation. These results suggest that TF and TF-enriched extracts stabilize skin structure by inducing subcutaneous fat production (Figure 5).

## 3. Discussion

Tea is a widely consumed drink around the world that is a dietary source of bioactive compounds with numerous health benefits [29]. Various types of tea can be produced via different manufacturing processes, such as non-fermented (e.g., GT), semi-fermented (e.g., oolong tea), and fermented (e.g., black tea) [30]. Fermentation or extraction modifies chemical components in tea, which can alter their medicinal effects [30,31]. Polyphenolic compounds in tea are of particular interest due to their antioxidant and anti-obesity activities [32,33]; indeed, many previous studies have demonstrated the health benefits of polyphenol-enriched tea extracts [34,35,36]. However, few studies to date have focused on the individual chemical components such as TF, TF3G, TF3′G, and TFDG (Appendix A), especially in the context of lipid metabolism. Possible reasons for this limited research are as follows: (1) it is easier to evaluate the effects of the whole extract rather than individual components; (2) effective separation of individual components is technically challenging; and (3) fat cells are derived from visceral tissues that benefit from reduction. In this study, we investigated the effects of TF on subcutaneous adipocytes since these cells play an important role in lipid metabolism beneath the dermal layer and are a potential target for anti-aging products.

Adipocytes of murine origin (e.g., 3T3-L1 and 3T3-F442A cells) have been widely used as a model system since they can form LDs within 14 days after differentiation. However, in this study we used hSCF cells as a model of subcutaneous fat cells despite their long period of differentiation, since they are derived from human tissue. hSCF cells required approximately three weeks for differentiation, which was observed at a rate of 60–80%. LDs formed in the differentiated hSCF cells, accompanied by upregulation of lipogenesis-related gene expression (Figure 3A, left panel and Figure 4A).

The effect of CoF-GT on hSCFs differed markedly from that of conventional polyphenols. Unlike F-GT, CoF-GT stimulated lipogenesis in hSCF cells; this effect is presumably distinct from its anti-obesity effects such as lipid disruption or suppression of differentiation (Figure 3). The treatment of F-GT, which increased overall polyphenol content, indicates that TF content ratio in the extract is a critical determinant of LD formation in hSCF cells (Figure 3B, lower panel and Figure 3C).

The hSCFs used in this experiment are responsible for the structure of human skin despite being fat-producing adipocytes, and function as brown adipose tissue to maintain thermogenesis [37,38,39]. As such, hSCFs are more suitable for anti-skin aging experiments than cells derived from white adipose tissue in visceral fat, which increases with age and is a site of inflammation [40,41]. Cellular changes caused by TF or TFDG treatment were accompanied by altered expression of lipogenesis-related genes and adipokine release (Figure 4). In particular, increased level of PPARγ could stimulate adiponectin with anti-aging properties through inhibiting destruction of extracellular matrix (e.g., type 1 collagen and elastin) in skin [41,42]. The opposing effects of CoF- and F-GT on hSCF cell differentiation are due to the ratio of TF contents. CoF-GT, which has an exceptionally high TF content, stimulates lipogenesis and the formation of LDs (Figure 5).

## 4. Materials and Methods

### 4.1. Reagents and Materials

Fresh C. sinensis leaves were harvested between August and September 2017 in Jeju, Korea. *C. coronarium* L. was purchased from a market in Kyungdong, Korea. TFs were from Wako Pure Chemical Industries (Osaka, Japan). Acetonitrile and methanol for chromatography were from Thermo Fisher Scientific (Waltham, MA, USA). Water for analytical high-performance liquid chromatography (HPLC) was from Burdick and Jackson (Morris Plains, NJ, USA). Ultra-pure deionized water for preparative HPLC (18.2 MΩ·cm) was prepared from a Direct-Q system (Merck, Darmstadt, Germany). Dimethylsulfoxide (DMSO) and formic acid were from Sigma-Aldrich (St. Louis, MO, USA). All other chemicals were of analytical grade or higrer.

### 4.2. Production of TF-Enriched CoF-GT and Conventionally Fermented F-GT Fractions

Fresh GT and GC leaves were washed twice with deionized water and excess water was removed by light tapping. The leaves were soaked in liquid nitrogen and then crushed into a fine powder that was stored at −80 °C. CoF- and F-GT were prepared as follows. CoF-GT was fermented from a mixture of fresh GT powder (100 g) and frozen fresh GC (50 mg). F-GT was also fermented from frozen fresh GT powder (100 g) in a thermal jacket z-blade mixer (IKA, Staufen im Breisgau, Germany) at 37.5 °C. After 3 h incubation, the mixture was extracted with 70% ethyl alcohol (*v*/*v*) for 2 h. Solid particles/debrises were removed by passage through a 90-mesh sieve followed by a 0.22 μm pore filter (Dow Corning, Corning, NY, USA). Filtered samples were evaporated with a Hei-VAP Rotary Evaporator (Heidolph Instruments, Schwabach, Germany) and then powdered using a FreeZone freeze dryer (Labconco, Kansas City, MO, USA).

### 4.3. HPLC Analysis of CoF- and F-GT Fractions

Each powered extract (8 g) was dissolved in a mixture of 100 mL DMSO/methanol/ethanol (5:45:50, *v*/*v*) with 30 min sonication and then centrifuged, followed by filtration through a 0.45 μm polytetrafluoroethylene syringe filter (Pall Corp., Port Washington, NY, USA). An 82 mL volume of filtered sample was injected into ÄKTApurifier 10 (GE Healthcare, Stockholm, Sweden) equipped with a 50 mL sample loop and a photo diode array detector at 275 and 365 nm. Preparative separation was performed with an AQ-HG octadecylsilyl column (120 Å, 10 μm, 20 × 250 mm, column volume = 78.5 mL) (YMC Co., Kyoto, Japan). Gradient elution was carried out with pure water (solvent A) and acetonitrile (solvent B). The flow rate of the mobile phase was 10 mL/min with an injection volume of 8.2 mL. All solvents were filtered, degassed, and maintained under pressure. Fractions (110 mL) were collected after sample injection. Four fractions were prepared for each cycle and collected in separate bottles over 10 injection cycles (Appendix A). Every fourth fraction was evaporated in a Hei-VAP Rotary Evaporator (Heidolph Instruments), freeze dried, and stored at −20 °C until analysis.

### 4.4. Cell Culture and Differentiation

hSCF cells and subcutaneous pre-adipocyte medium were purchased from ZenBio (Research Triangle Park, NC, USA). The cells were cultured in a humidified 5% CO_2_ incubator. To induce differentiation, the cells were cultured in Dulbecco’s modified Eagle’s medium (Lonza, Walkersville, MD, USA) containing 10% fetal bovine serum (PAA, Pasching, Austria) along with 10 µg/mL insulin, 0.5 mM 3-isobutyl-1-methylxanthine, 1 µM dexamethasone, and 1 µM troglitazone (all from Sigma-Aldrich) for 21 days. The medium was replaced every other day.

### 4.5. Cell Viability Assay

The viability of hSCF cells was measured with the EZ-Cytox assay kit (Daeil Lab Service, Seoul, Korea) according to the manufacturer’s instructions. Briefly, cells were cultured for 7 days and treated with various concentrations of test substance for 24 and 72 h. EZ-Cytox solution (10 μL) was added to each well followed by incubation at 37 °C for 2 h. Absorbance at 450 nm was measured with a spectrophotometer (Synergy H2; BioTek, Winooski, VT, USA). The experiment was performed in triplicate and data are presented as the absorbance value.

### 4.6. Analysis of Triglyceride (TG) Content by Oil Red O (ORO) Staining

Differentiated hSCF cells were washed twice with cold phosphate-buffered saline (PBS) and fixed with 3.7% formaldehyde (Sigma-Aldrich) for 1 h. The fixed cells were washed with 60% propylene glycol (Sigma-Aldrich) in PBS and then stained with a working solution of Oil Red O (0.7% ORO stock in 60% propylene glycol; Sigma-Aldrich) for 30 min. The cells were washed three times with 85% propylene glycol and rinsed with tap water. Lipid droplets (LDs) stained with ORO dye were visualized with an IX71 microscope (Olympus, Tokyo, Japan).

To quantify LDs, stained cells were washed with 70% ethyl alcohol (Sigma-Aldrich) and the triglyceride (TG)-bound ORO was extracted with 4% Nonidet P-40 (Sigma-Aldrich) in isopropyl alcohol for 20 min. The absorbance of the extract at 490 nm was measured with a spectrophotometer.

### 4.7. Real-Time Quantitative Plymerase Chain Reaction (RT-qPCR)

Total RNA was extracted using TRIzol reagent (Life Technologies, Carlsbad, CA, USA) according to the manufacturer’s instructions, and cDNA was synthesized from approximately 1 μg total RNA using a reverse transcription kit (Promega, Madison, WI, USA). The cDNA was used as a template for RT-qPCR amplification on a 7500 Fast Real-Time PCR System (Life Technologies) using the following TaqMan probes: peroxisome proliferator activated receptor gamma (PPARγ) (#Hs01115513_m1), adiponectin (ADIPOQ; #Hs00605917_m1), and glyceraldehyde 3-phosphate dehydrogenase (GAPDH; #4352339E). Data were obtained from three independent experiments and are presented as fold change relative to the GAPDH level in the sample.

### 4.8. Enzyme-Linked Immunosorbent Assay (ELISA) for Secreted Adiponectin

hSCFs were treated with various concentrations of TG, TFDG, CoF-GT, and F-GT and differentiated for 21 days. The culture medium was collected and centrifuged at 13,000 rpm for 15 min to remove any debris. Secreted adiponectin level was measured by ELISA using a commercial kit (Enzo Life Sciences, Farmingdale, NY, USA) according to the manufacturer’s instructions.

### 4.9. Statistical Analysis

Data are presented as mean ± SD. One- or two-way ANOVA analysis of variance were used to analyze differences between two and multiple groups, respectively. The threshold for statistical significance was set at *p* < 0.05.

## 5. Conclusions

The results of this study demonstrate that TF and TF-enriched CoF-GT plays an important role in hSCF cell differentiation due to different biological effects depending on cell types whereas TFDG and conventionally fermented GT inhibits LD synthesis. Our findings suggest that with rapid fermentation and effective trans-dermal delivery, TF and TF-enriched CoF-GT can be an effective agent for preventing human skin aging.

## Figures and Tables

**Figure 1 ijms-20-02034-f001:**
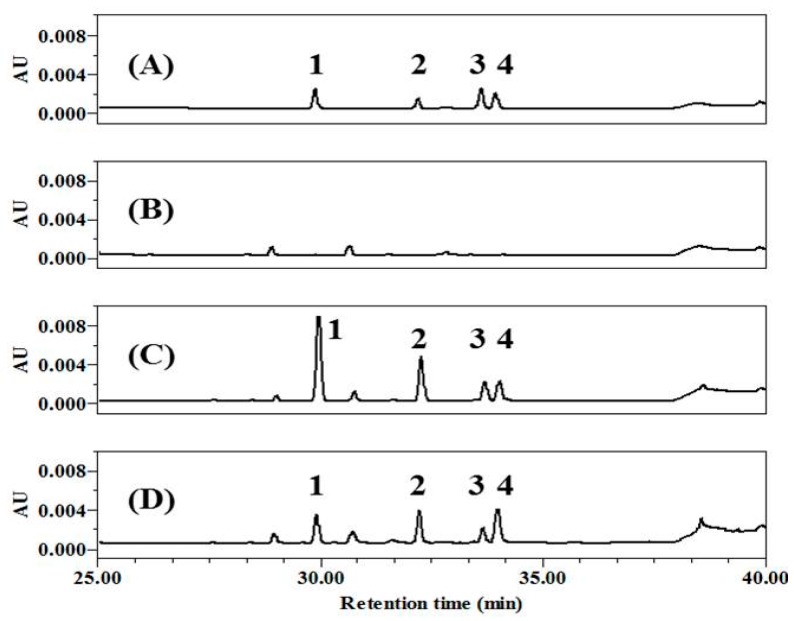
High-performance liquid chromatography (HPLC) analysis of theaflavins (TFs). (**A**) Standard chemicals; (**B**) Green tea extract; (**C**) co-fermented green tea (CoF-GT) fraction; (**D**) fermented green tea (F-GT) fraction. Data were monitored at 270 nm and compounds were assigned as follows: Peak 1, theaflavin (TF); Peak 2, theaflavin-3-gallate (TF3G); Peak 3, theaflavin-3′-gallate (TF3′G); and Peak 4, theaflavin-3,3′-digallate (TFDG).

**Figure 2 ijms-20-02034-f002:**
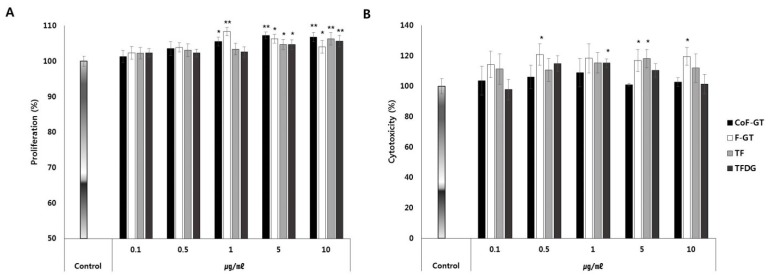
Effect of TF on human subcutaneous fat cells (hSCFs) viability. (**A**,**B**) Proliferation (**A**) and cytotoxicity (**B**) were evaluated 24 and 72 h after treatment, respectively. Data are presented as mean ± standard deviation (SD). * *p* < 0.05, ** *p* < 0.01.

**Figure 3 ijms-20-02034-f003:**
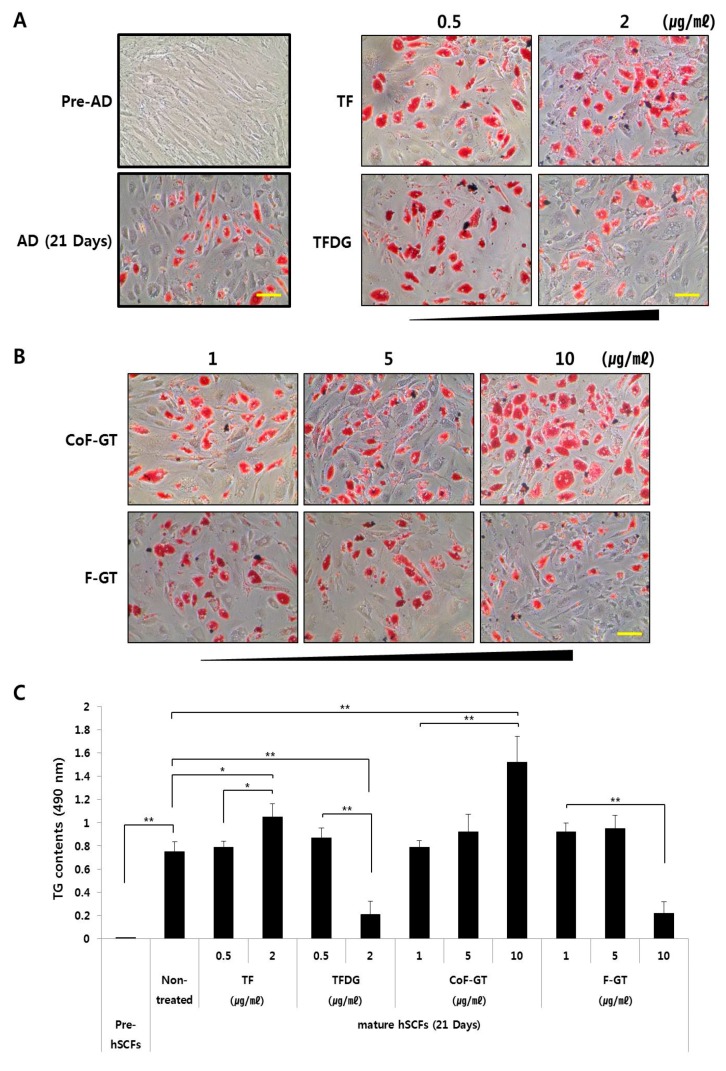
Effect of TF on hSCF differentiation. Pre- and mature adipocyte states of hSCFs fixed and stained with Oil Red O (ORO) at indicated time points and examined by light microscopy. (**A**) **Left**, untreated group; **right**, TF- and TFDG-treated groups used as a positive control; (**B**) CoF- and F-GT treated groups. Scale bar, 200 μm; (**C**) TG content was determined by measuring the absorbance at 490 nm. Data are presented as mean ± SD. * *p* < 0.05, ** *p* < 0.01.

**Figure 4 ijms-20-02034-f004:**
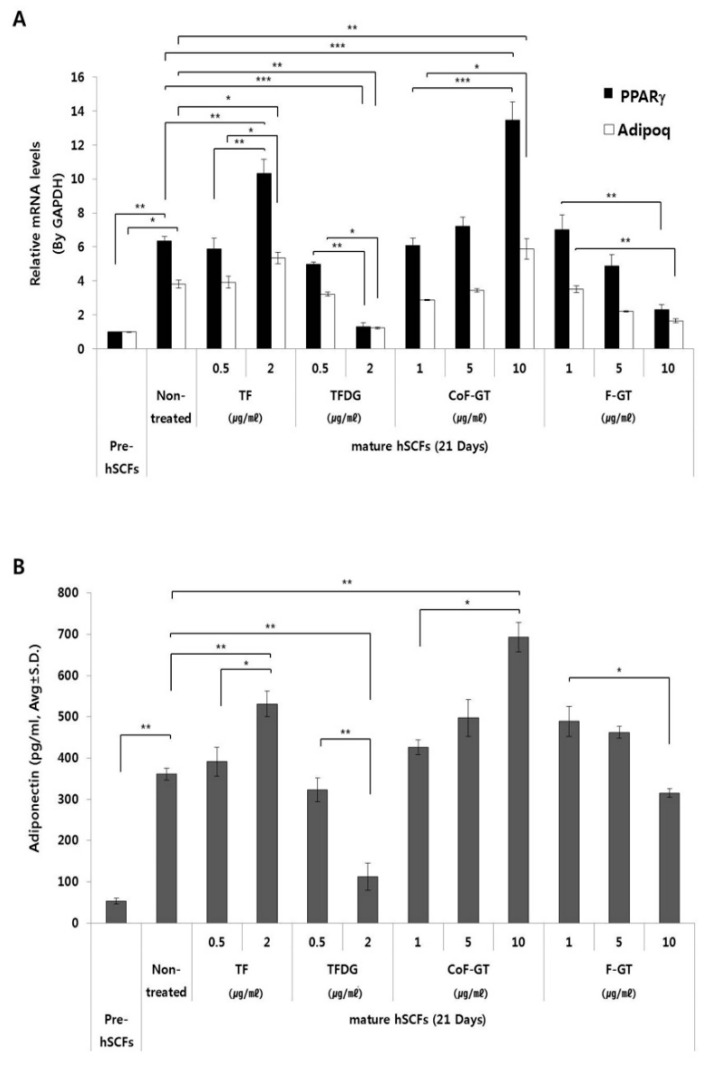
Changes in the expression of adipogenesis-related genes and quantitative analysis of adiponectin release into culture medium. (**A**) After 21 days of hSCF differentiation, *peroxisome proliferator activated receptor gamma* (PPARγ) and *adiponectin* (ADIPOQ) mRNA expression levels were evaluated by real-time quantitative polyperase chain reaction (RT-qPCR); (**B**) Adiponectin concentration in cell culture supernatant after 21 days of differentiation was measured by enzyme linked immunosorbent assays (ELISA). Data are presented as mean ± SD. * *p* < 0.05, ** *p* < 0.01, *** *p* < 0.001.

**Figure 5 ijms-20-02034-f005:**
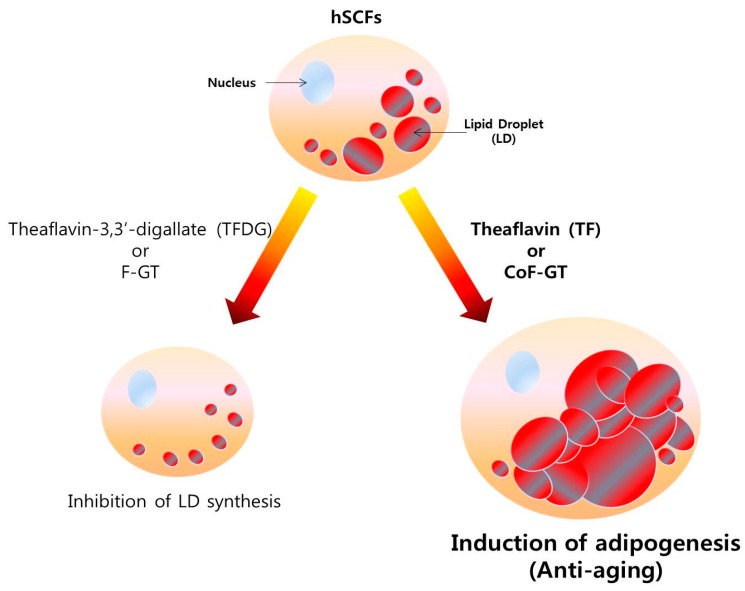
Function model of CoF- and F-GT fraction in hSCFs differentiation. Conventionally fermented F-GT fraction decreases the number of LDs while TF-enriched CoF-GT fraction induces their production in hSCFs.

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
