# Peer review of "Theaflavin-Enriched Fraction Stimulates Adipogenesis in Human Subcutaneous Fat Cells"

_ijms, 2019, doi:10.3390/ijms20082034_

Reviewer 1 Report

Authors enriched Theaflavin in green tea and adipogenesis stimulation was assessed. Manuscript is clear and easy to read.

However, statistical analysis may be concern for this manuscript. 

Authors mentioned t-test or anova. But, there are no one-one comparison should be conducted for given figures. Only one way or two way ANOVA should be done. Please specify which statistical analysis was conducted for all figures.

for example....

Figure 2 data about proliferation and cytotoxicity has several * with statistical significance. It was very unclear what kinds of statistical analysis was conducted (t-test or anova). Significant differences against what data point(s), Please clarify in the manuscript.

Author Response

We thank reviewer’s comment. As suggested, we revised the statistical analysis in Materials and Methods (line 274-275). One or two-way ANOVA test was used to determine whether there are any statistically significant differences between groups in all experiments. In detail, one-way ANOVA was used in cell viability test (Figure 2), whereas two-way ANOVA was used in differentiation test (Figure 3 & 4). We removed the two-tailed Student’s t-test instead after checking our data. Sorry for the hesitation.

We highlighted the revised sentence in the manuscript as follows, which provides a better understanding of data (Line 278-279).

; Data are presented as mean ± SD. One- or two-way ANOVA analysis of variance were used to analyze differences between two and multiple groups, respectively. The threshold for statistical significance was set at P < 0.05.

Reviewer 2 Report

The manuscript demonstrates the effect of theaflavin (TF) in green tea (GT) on skin rejuvenation effects. The authors have used primary human subcutaneous fat cells (hSCF) to examine the ratio of TF contents in inducing subcutaneous adipogenesis supporting skin structure by modulating lipid metabolism. The authors have evaluated adipogenesis by lipid droplet (LD) formation visualized by Oil Red O staining, analyzing adipogenesis-related factors by real-time quantitative PCR; and measuring the concentration of adiponectin released into the culture by ELISA. This is a  interesting study and provides information that TF and TF-enriched co-fermented-GT plays an important role in hSCF cell differentiation having different biological effects and may be effective agents for prevention of skin aging. Overall, the study is well-intentioned and presented uses methods, which are straightforward, with which the authors have much experience. There are few suggestions to improve the quality of the manuscript.

1.       The authors should avoid too much use of abbreviations in the manuscript.

2.       Rationale for using lipid droplet formation assay should be provided.

3.       The authors should expand the discussion by adding information about PPARgamma and adiponectin and their role in skin aging.

Author Response

1. We thank the reviewer’s comment. As the reviewer mentioned, we also agree that excess use of abbreviation should be avoided. We reduced unnecessary abbreviation such as ELISA, LD and ORO because they are generally known as well as are not used that much. However, we still used ADIPOQ and APN because they could be confused.
2. We appreciate for reviewer’s comment. As mentioned, we also think that it is important to explain the rationale for using lipid droplet formation assay for a better understanding of our manuscript. According to previous literatures, lipid droplet formation is closely related to adipocyte differentiation (NagayamaM.;Uchida T.;Gohara K. Temporal and spatial variations of lipid droplets during adipocyte division and differentiation. J. Lipid Res. 2007, 48, 9-18, Ducharme N.A.; Bickel P.E. Mini-review: lipid droplets in lipogenesis and lipolysis. Endocrinology 2008, 149(3), 942-949.). During adipocyte differentiation, lipid droplets are formed and changed in aspects of size and shape.

Therefore, we added highlighted the following sentences about the rationale for using lipid droplet formation assay in Section 2.3. (Line 101-104).

; The differentiation of hSCF cells was confirmed by the formation of intracellular lipid droplets after 21 days of culture (Figure 3A, lower left panel) because lipid droplet formation is closely related to adipocyte differentiation and its observation could be useful for monitoring the differentiation.

3. We also agree with reviewer’s comment and suggestion. To best of our knowledge, increased mRNA and protein expression level of PPARγ could stimulate the differentiation of adipose tissues and adipocyte specific cytokine (adipokine): particularly, adiponectin, mainly expressed in adipose tissues. According to the previous study, Noh group mentioned that adiponectin is stimulated in mature adipocytes by induction of adipogenesis-related genes such as PPARγ and CCAAT/enhancer binding proteins (C/EBP). Also, increased adiponectin enhances the anti-aging properties through inhibiting destruction of extracellular matrix (e.g., type 1 collagen and elastin) in skin [Reference #43]. In light of this, increased adiponectin plays an important role in skin aging. We mentioned highlighted such information in the Discussion part (Line 200-202).